# Disease Diagnosis Using Retinal Vasculature: Insights from Flammer Syndrome and AI

**DOI:** 10.3390/brainsci15090919

**Published:** 2025-08-26

**Authors:** George Ayoub

**Affiliations:** Psychology Department, Santa Barbara City College, Santa Barbara, CA 93109, USA; neuro@sbcc.edu

**Keywords:** Flammer Syndrome, machine learning, AI, lifestyle medicine, glaucoma, AMD, lymphedema, vitamin therapy

## Abstract

The retinal vasculature provides a unique and non-invasive window into the health of the circulatory system. Josef Flammer, a pioneer in ocular vascular research, was the first to systematically describe how the state of retinal blood vessels reflects broader cardiovascular health. Because the retina is the only part of the human body where blood vessels can be visualized non-invasively, it serves as a valuable proxy for understanding microvascular conditions elsewhere, including the heart, brain, and lymphatics. Recent work has shown that retinal vasculature can be used as a proxy for microcirculatory dysfunction in other body systems, and that treatment using medical doses of vitamins can restore microcirculation, easing symptoms in disorders as diverse as glaucoma, AMD, and lymphedema without the need of pharmacological agents. The advent of machine learning tools to read retinal images promises both early detection of conditions and simplified monitoring of treatment progression.

## 1. Retinal Vasculature Reflects Cardiac Health

Flammer pioneered the concept that microvascular abnormalities in the retina are often the earliest indicators of systemic vascular dysfunction. The guiding principles of his research recognized that systemic vascular dysregulation, driven by oxidative stress, endothelial dysfunction, and autonomic imbalance, manifests similarly in retinal and cardiac vasculature [1,2].

Microvascular irregularities such as vessel narrowing, increased tortuosity, or dysregulated autoregulation in the retina may correlate with similar pathologies in cardiac vasculature. This concept has been supported by studies showing that retinal vascular caliber and integrity are associated with risks of hypertension, coronary artery disease, and stroke [1,3].

Flammer’s hypothesis was grounded in the understanding that systemic vascular dysregulation, influenced by factors such as oxidative stress, endothelial dysfunction, and autonomic imbalance, manifests similarly in both retinal and cardiac vessels. Thus, routine retinal imaging could potentially offer a non-invasive means to detect early cardiovascular abnormalities [1].

Clinical implications of Flammer’s work include using routine retinal imaging, such as fundus photography, OCT, and vessel analysis, as a non-invasive screening tool for preclinical cardiovascular diseases and for stratifying patient risk [4,5]. Retinal assessment was integrated with systemic assessments, highlighting correlations with low blood pressure, altered cardiac autonomic function, and increased risk of angiopathic events including stroke and myocardial ischemia [1].

By establishing links between retinal vessel changes and cardiovascular conditions, Flammer advanced the use of retinal biomarkers in personalized medicine and preventive vascular care [1,2].

Table 1 provides a list of the retinal variations that are observed and what disease states are associated with each of these changes, as well as the clinical implications from these observations.

## 2. Flammer Syndrome: A Diagnostic Window into Vascular Health

Flammer Syndrome (FS) is a clinical phenotype characterized by primary vascular dysregulation, which manifests through symptoms such as cold extremities, low blood pressure, increased sensitivity to stress, and migraine-like headaches, as shown in Table 2. It is not a disease but a predisposition that reflects an underlying vascular instability, particularly in response to stressors such as hypoxia or emotional stimuli. Flammer identified that individuals with Flammer Syndrome often exhibit retinal vascular dysregulation, including delayed autoregulation and vasospastic responses [11,12,13].

This syndrome has proven valuable as a diagnostic tool for identifying individuals at risk of vascular disorders, including normal-tension glaucoma, where ocular perfusion pressure is compromised despite normal intraocular pressure [14]. Diagnosis has typically been based on symptomatology and vascular response assessments, and it highlights the importance of personalized vascular profiles in managing both ocular and systemic diseases. Table 2 indicates common FS symptoms and their clinical correlates.

**Table 2 brainsci-15-00919-t002:** Flammer Syndrome symptoms and their clinical implications.

Symptom	Clinical Implication
Cold extremities	Indicator of vascular dysregulation [15]
Low blood pressure	Systemic vascular instability [15]
Stiff/irregular retinal vessels	Marker for disturbed ocular blood flow regulation [15,16]
Reduced flicker response	Evidence of retinal endothelial dysfunction [16]
Increased retinal venous pressure	Associated with risk for glaucoma and vascular events [1,16]

There are some molecular and systemic biomarkers for FS. Subjects with FS often have elevated levels of plasma endothelin-1, a potent vasoconstrictor, contributing to their vascular dysregulation [1,15]. Subjects also often have elevated plasma homocysteine levels, and when these levels are reduced the impact from FS is also reduced [14,17]. These biomarkers may provide useful indicators to assess the progress of clinical therapy.

While many with FS remain healthy, the condition is a recognized risk factor for specific ocular and systemic diseases. One such disease is normal-tension glaucoma. FS is overrepresented in these patients, who often experience ocular perfusion deficits without elevated intraocular pressure. Other conditions include retinal vein occlusions, anterior ischemic optic neuropathy, and central serous chorioretinopathy, where the incidence is increased among those with FS.

Additionally, there is an association of FS with other disorders. Multiple sclerosis, sudden cardiac ischemia, migraine, and even hereditary neuropathies show higher prevalence in individuals with FS [1,15,16].

The early identification of FS has direct implications for preventive medicine and risk stratification. Retinal imaging and assessment for vascular dysregulation (e.g., flicker-light response tests) are valuable for identifying individuals at risk for glaucoma or other vascular disorders, often before conventional risk factors are apparent [15,16].

## 3. Retina and Vascular Disease

The retina is the only location in the human body where blood vessels can be directly and non-invasively visualized in vivo. This unique anatomical feature allows clinicians and researchers to study the microvascular system in detail and to detect early signs of both systemic and ocular diseases.

The retinal vasculature plays a critical role in ocular diseases such as glaucoma, age-related macular degeneration (AMD), and diabetic retinopathy. In glaucoma, reduced blood flow and vascular dysregulation contribute to optic nerve damage [16,18,19]. As Flammer explains in his detailed overview of ocular blood flow [20], retinal vascular dysregulation, particularly in individuals with Flammer Syndrome, leads to unstable ocular perfusion. This instability can cause oxidative stress, mitochondrial dysfunction, and ultimately the apoptosis of retinal ganglion cells. This is further supported by the studies showing the link between Flammer Syndrome and normal tension glaucoma, with evidence that in this condition there is an uncontrolled increase in retinal venous pressure and concomitant decrease in ocular blood flow giving rise to the glaucomatous condition [2,14,21].

Similarly, in AMD, compromised choroidal and retinal circulations play a key role in disease progression. Factors such as hypoxia, chronic inflammation, and impaired clearance of metabolic waste exacerbate retinal degeneration. This is supported by work identifying that oxidative stress and mitochondrial dysfunction are critical contributors to glaucoma pathogenesis [22]. In the case of AMD, recent findings underscore the role of vascular inflammation and lipid accumulation in drusen formation, which are essential contributors to photoreceptor degeneration and geographic atrophy [23,24].

Additionally, the patterns of retinal vascular alterations, such as arteriolar narrowing, venular widening, increased vessel tortuosity, and rarefaction, correlate strongly with several cardiovascular and systemic disorders, including hypertension, atherosclerosis, and diabetes mellitus, providing a non-invasive window to systemic disease detection [25,26]. These systemic disorder indicators are summarized in Table 3.

## 4. Novel Therapeutic Approaches in AMD and Glaucoma

Josifova et al. (2025) proposed a novel therapeutic approach targeting vascular inflammation and mitochondrial resilience in AMD [17]. This treatment (Ocufolin forte, Aprofol AG) combines selective anti-inflammatory agents with mitochondrial stabilizers to reduce reactive oxygen species and restore endothelial function in choroidal vessels. The treatment demonstrates potential to slow the progression of atrophy in AMD patients, particularly when initiated during early stages of the disease. Of note, Josifova et al. found that when Ocufolin forte was added to standard AMD therapy, there was a larger reduction in retinal venous pressure (RVP) and homocysteine, resulting in a decreased frequency of anti-VEGF injections [17].

Retinal venous pressure (RVP) is an emerging, non-invasive biomarker reflecting the health of the retinal and by extension systemic circulation. RVP closely parallels intraocular pressure in healthy individuals, but can become markedly elevated in disease states, including glaucoma, diabetic retinopathy, retinal vein occlusion, and conditions characterized by vascular dysregulation, such as Flammer Syndrome [1]. Elevated RVP often indicates local venous constriction mediated by vasoactive substances (notably endothelin-1), hypoxia, or inflammation, and it is associated with microvascular dysfunction and increased risk for hypoxic injury, retinal edema, and vein occlusion [1,27].

Homocysteine is a non-essential amino acid where elevated plasma levels promote vascular dysfunction through oxidative stress, endothelial damage, and pro-inflammatory mechanisms [27,28,29]. Mechanistically, homocysteine induces vascular endothelial dysfunction and promotes oxidative injury, thereby propagating microvascular pathology throughout the retina and choroid [28,30]

Similarly, Devogelaere & Schötzau (2021) [14] observed that RVP and homocysteine are elevated in both high tension and normal tension glaucoma. Significantly, using the same vitamin supplementation as Josifova et al. lowered RVP and homocysteine, pointing to a new treatment modality for glaucoma [14,31].

## 5. Mechanism of Action

Modern therapy for retinal diseases such as age-related macular degeneration (AMD) and glaucoma increasingly leverages integrated strategies that address underlying pathophysiological mechanisms rather than focusing solely on symptomatic relief. Notably, interventions that combine L-methylfolate, B vitamins, and antioxidants (such as with Ocufolin^®^ forte) employ a dual-action mechanism targeting both vascular modulation and mitochondrial protection [14,17]. The responses are summarized in Table 4.

### 5.1. Vascular Modulation

The therapy inhibits key pro-inflammatory cytokines and cellular adhesion molecules, reducing vascular leakage and leukocyte infiltration in retinal capillaries. This alleviates local inflammation, improves microvascular perfusion, and reduces ischemic damage to neural and vascular retinal tissue [17,32].

### 5.2. Mitochondrial Protection

The therapy includes agents that enhance mitochondrial membrane integrity, reduce oxidative phosphorylation stress, and promote mitophagy. This preserves photoreceptor viability and retinal pigment epithelial cell function, which are essential for maintaining retinal structure and function in AMD [17,33].

This integrated approach marks a paradigm shift from symptomatic management toward precision interventions tailored to addressing underlying pathophysiological mechanisms in vascular and mitochondrial health. The addition of mitochondrial stabilizers and anti-inflammatory components to vascular therapy not only slows progression of retinal atrophy but also provides broad neuroprotection [34]. Current and emerging data support this dual mechanism as a foundation for new standards in the management of AMD, glaucoma, and other vascular–retinal diseases [14,33,35,36].

**Table 4 brainsci-15-00919-t004:** Response to vitamin intervention such as Ocufolin forte.

Mechanism	Clinical Benefit	Reference
Vascular modulation	Reduced leakage, improved perfusion, less ischemia	[14,17,32]
Anti-inflammatory action	Lower cytokine/adhesion molecule levels, stabilized vessel barrier	[14,17,32,33]
RVP/homocysteine lowering	Decreased risk of edema, occlusion, and neural injury	[14,27,33]
Mitochondrial protection	Enhanced RPE/photoreceptor survival; delayed atrophy	[35]
Reduced oxidative stress	Attenuated disease progression	[35,36]

## 6. Endothelial Cell Health and B Vitamins

Retinal endothelial cells form the inner lining of retinal blood vessels. They serve multiple functions beyond the transport of blood. Specifically, they maintain the blood–retina barrier, which regulates nutrient and molecular exchange while blocking pathogens and toxins; they support vascular homeostasis, ensuring the retina receives sufficient oxygen and nutrients; and they produce substances that mediate blood coagulation, vascular tension, and new vessel growth important for tissue repair and adaptation [37,38].

In both systemic and ocular disease states, endothelial cell dysfunction is a fundamental driver of pathology. When endothelial cells are injured or have reduced function, this leads to several functional deficiencies. These damages include an increased permeability and barrier breakdown. This initiates fluid leakage, macular edema, and risk of hemorrhage, as seen in diabetic retinopathy and other vasculopathies [37,38,39]. There can also be an impaired regulation of vascular tone. This is when a loss of normal endothelial function leads to abnormal vessel constriction or dilation, tissue ischemia, and hypoxic injury. Another complication is chronic inflammation and leukocyte adhesion. In this case, elevated cytokines (such as IL-6, IL-8) and adhesive molecules at the endothelium perpetuate low-grade vascular inflammation, exacerbating vessel occlusion and ischemia [38]. Excessive neovascularization can also occur. This is when there is pathological angiogenesis, driven by overproduction of VEGF and other growth factors, which underlies advanced diabetic retinopathy and age-related macular degeneration [38]. And often there is oxidative stress. The high metabolic demand in the retina makes endothelial cells susceptible to oxidative damage, further impairing their barrier and regulatory functions [40]. These damages lead to the clinical consequences that are seen as a result of retinal disease, as shown in Table 5.

Emerging evidence indicates that supporting vascular health through B vitamin supplementation may yield significant benefits. Josifova (2025) highlights the protective effects of B vitamins in reducing homocysteine levels, enhancing endothelial function, and mitigating oxidative stress in retinal and choroidal vessels [17]. These effects not only support retinal health but may also extend to systemic vascular networks [41].

Importantly, this insight into vascular biology has broader implications for conditions such as lymphedema, where insufficient lymphatic drainage mirrors the challenges observed in venous insufficiency. As noted by Ayoub (2024), the structural and functional parallels between blood vasculature and lymphatic vessels suggest that interventions aimed at improving microvascular perfusion may also support lymphatic function [42]. In particular, B vitamin supplementation has shown promise in alleviating symptoms of lymphatic disorders, likely through enhanced endothelial resilience and reduced inflammatory signaling.

This highlights the critical importance of the metabolic and nutritional modulators of endothelial health. Elevated levels of homocysteine disrupt the endothelial barrier integrity, promote oxidative stress, and heighten risk for retinal and systemic vascular disease [40]. Adequate B vitamins, particularly B6, B9 (folate), and B12, are essential to the support of endothelial cell metabolism, reducing homocysteine, enhancing antioxidant defenses, and preserving microvascular function. These benefits are demonstrated in retinal, choroidal, and even lymphatic endothelium [14,17,42,43]. Additionally, genetic and age-related factors, such as a reduced absorption of natural B vitamins (especially methyl-folate) in older adults and those with MTHFR gene variants, can further compromise endothelial health, raising risk for ocular and systemic vascular disease [43].

Thus, what we learn from retinal vasculature may be generalizable to microcirculatory conditions, including both systemic and lymphatic dysfunction. The integration of micronutrient therapies, particularly B vitamins, represents a promising direction for addressing both vascular and lymphatic insufficiency through shared biological pathways. In this regard, a new recent study reveals that primary lymphedema may be a disorder due to insufficiency of nutrients for endothelial cell health, reducing lymphatic drainage, and responding well to medical doses of natural B vitamins [42].

These treatments, with both retinal and lymphatic veins responding to identical therapeutic medical dosage (each study used the same medical food, Ocufolin, which is an AREDS2+ therapeutic, as the source of natural B vitamins) open an avenue to possible therapeutics to stave off neurodegenerative disorders such as dementia and Parkinson’s disease. We may be on the cusp of identifying that symptoms in such neurodegenerative disorders are at least in part reliant on endothelial cell health. Given that vitamin absorption often declines with age and the abundance of OTC vitamin supplements use synthetic folic acid rather than natural methyl-folate, it is likely that reduced levels of natural Vitamins B9 and B12 become less available due to competition from elevated levels of oxidized folic acid, which slows down cellular absorption of natural B9 and is responsible for other vascular and developmental challenges [44].

Thus, endothelial cell health is pivotal in the prevention and management of retinal and systemic vascular diseases. Dysregulation at this cellular level initiates a cascade culminating in vision-threatening outcomes. Addressing modifiable factors, such as micronutrient deficiencies and oxidative stress, offers promising new directions for therapy, with B vitamin supplementation and regulation of homocysteine at the forefront of emerging evidence-based approaches. This is summarized in Figure 1.

## 7. AI and Retinal Fundoscopy: Detecting Glaucoma and Autism

Recent advances in machine learning and artificial intelligence (AI) have revolutionized the interpretation of retinal images and opened opportunities for using such images in disease diagnosis [45,46,47,48]. The systems have used fundus images, OCT (Optical Coherence Tomography) scans, and visual field tests, and developed systems that take these images and use one of several ophthalmological algorithms (such as searches of optic disc variations) [49,50] or deep learning where the system entrains on images pre-sorted as healthy or glaucomatous and is programmed to choose images similar to the diseased ones in novel images [51,52,53]. These deep learning systems can use convolutional neural networks to process vascular image features and/or an artificial neural network to analyze retinal thickness parameters determined from OCT images [50,53,54]. These systems have shown accuracy rates consistent with trained ophthalmological teams separately assessing the same images for glaucoma [47,48].

Benny Zee and colleagues have demonstrated that a machine learning system applied to retinal fundus images can identify not only ocular diseases such as glaucoma [55] but also neurodevelopmental conditions like autism [56]. In recent studies, Zee et al. developed a machine learning-based retinal image analysis system that could detect diabetic retinopathy and glaucomatous changes with high sensitivity and specificity [14,31].

Further expanding the potential of this technology, Zee’s team published findings indicating that their AI could identify children that were diagnosed as autism spectrum disorder (ASD) based on unique retinal vascular patterns [56,57]. This suggests that subtle neurovascular signatures in the retina may reflect broader neurodevelopmental anomalies, opening new frontiers for early screening and diagnosis of retinal and neurodevelopmental disorders. The diagnostic value of such a non-invasive screen for a wide variety of conditions gives hope that in the future, such screenings can be deployed early and at a local level to address these conditions at their earliest stages. In this regard, a recent report has shown that a measure of retinal vasculature can provide early detection of Alzheimer’s dementia decades before symptoms are evident [58]. It thus seems likely that AI readings of retinal images will in the near future identify those at higher risk for multiple conditions and allow active monitoring of treatment success.

In addition to using AI with fundoscopic images for diagnosis of conditions, a recent publication used movement variations. By having participants wear a sensor glove, this study found that an AI system could be trained to assess whether each person was neurotypical, had ASD or ADHD, or both ASD and ADHD [59]. While this work is preliminary and the accuracy is a work in progress, it illustrates that using quantifiable body features may be a useful adjunct in making diagnoses.

One caveat is whether there is similarity in retinal images based on ethnographic origins. Studies to date have found that while there is a strong association between retinal pigmentation and self-reported ethnicity, there is substantial overlap in pigmentation scores among different ethnic groups. This means that the Retinal Pigment Score has been developed to quantify pigmentation from fundus photographs, and this score is influenced by genetic loci associated with skin, iris, and hair color. However, the distribution of RPS overlaps significantly across ethnic groups, indicating that family of origin (genetic background) plays a more direct role than broad racial or ethnic categories [60]. In addition, certain structural features of the retina, such as the pattern of retinal blood vessel arborization (the way blood vessels branch), can show familial similarities and may be genetically transmitted. However, the father away from the optic nerve head, the more individualized the patterns are, suggesting both inherited and acquired influences [61]. Thus, the variations by disease may provide a greater indicator than variations by genetics. Additionally, AI diagnostics can be debiased with the use of synthetic fundus images, as has been demonstrated for diabetic retinopathy [62].

Given the Josifova, Devogeleare, and Ayoub findings that such diverse conditions can effectively be reversed with a nutritive remedy, it seems likely that with early diagnosis, lifestyle changes can be used to rectify the nutritive deficiencies underlying symptoms that develop in such conditions. One can readily envision a diagnosis used to prescribe a medical dose vitamin treatment to address the immediacy of the disorder while introducing lifestyle changes to diet to ensure appropriate level of the key vitamins are incorporated daily in their natural form and reducing synthetic forms of the vitamins from consumption [44]. This dietary solution requires maintaining a healthy microbiome to enable processing of these nutrients into the absorbable, active forms utilized by the body.

## 8. Conclusions

From the foundational insights of Flammer to cutting-edge AI-driven diagnostics, the study of retinal vasculature has emerged as a powerful tool for understanding both systemic and ocular health. The retina, in its accessibility and vascular complexity, offers a unique vantage point to monitor cardiovascular function, predict ocular diseases like glaucoma and AMD, and even detect neurodevelopmental disorders such as autism. As imaging technologies and machine learning algorithms continue to evolve, the retina may become an increasingly central focus in both clinical and preventive medicine. Such early detection would permit facilitate therapy that treats underlying nutritional deficits that are at the root of each disease, well in advance of disease progression. Since such nutritional therapies are slower to act but prolonged in their efficacy [42], this can advance treatment of chronic conditions in a manner that provides a wide public good.

On can readily envision incorporation of fundoscopic images at annual exams to provide potentially as much guidance to disease onset and progression as has traditionally been provided by the history and lab tests used currently [63]. Such a diagnostic method also promises to democratize diagnosis, making it available in remote, rural, and underserved areas, collecting the fundoscopic images with a smart phone attachment.

## Figures and Tables

**Figure 1 brainsci-15-00919-f001:**
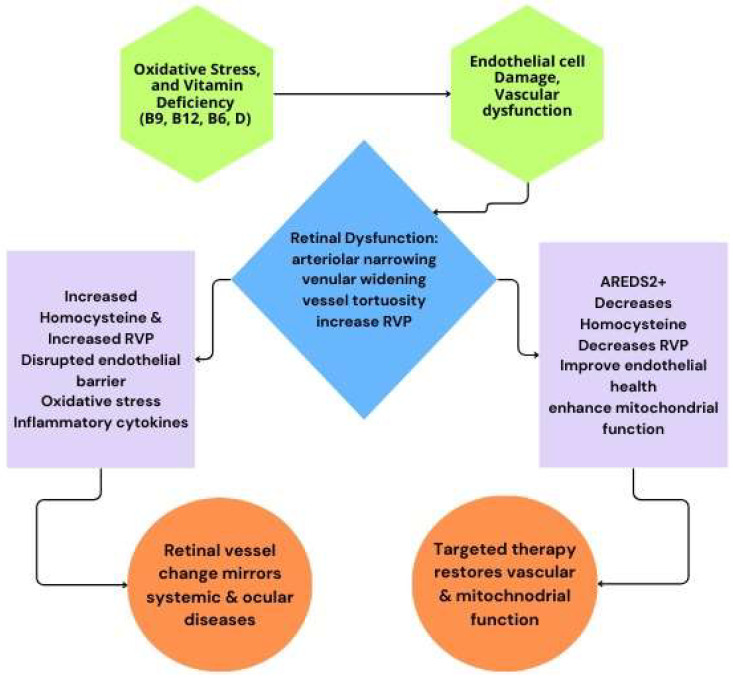
Summary schematic of the response to retinal vasculature dysfunction. Top (green hexagons) are the factors leading to endothelial cell damage, which creased vascular dysfunction, that leads to retinal dysfunction (blue diamond). The left box (lavender) is untreated, leading to systemic and ocular diseases (left orange circle). The right box (lavender) is treated with AREDS2+ therapy, leading to restored function (right orange circle). In either case, the retinal vasculature is changed, which is visualized in fundoscopic images, and allows diagnosis from these images of retinal and systemic dysfunction.

**Table 1 brainsci-15-00919-t001:** Retinal changes observed by Flammer and the diseases that are associated with these changes, along with clinical implications.

Retinal Change	Associated Diseases	Clinical Implication
Arteriolar narrowing	Hypertension, stroke, CAD *	Increased cardiovascular risk, indicative of systemic microvascular damage [5,6,7,8]
Venular widening	Stroke, heart failure, CAD *	Higher risk of incident stroke and heart disease [5,6,7]
Vessel tortuosity, rigidity	Flammer Syndrome, glaucoma	Impaired autoregulation, increased risk for vascular events [1,2,9]
Reduced capillary density	CAD *, peripheral artery disease	Lower vascular reserve, early marker of systemic atherosclerosis [1,4,10]
Abnormal arterial reflex/nicking	Hypertension, arteriosclerosis	Indicator of chronic vascular stress, predicts cerebrovascular events [5,7]

* CAD = coronary artery disease.

**Table 3 brainsci-15-00919-t003:** Systemic disorders and the observable retinal changes [22,23].

Disorder	Retinal Indications
Hypertension	Arteriolar narrowing, arteriovenous nicking (compressed vein), retinal hemorrhages and exudates
Diabetes mellitus	Microaneurysms, dot-blot hemorrhages, cotton wool spots, venous beading, neovascularization
Atherosclerosis	Increased vessel wall thickness, altered vessel caliber

**Table 5 brainsci-15-00919-t005:** Retinal Endothelial Dysfunction and Retinal Disease Effects.

Disease Consequences	Epithelial Dysfunction Manifestation	Clinical Consequences
Diabetic Retinopathy	Barrier loss, neovascularization, inflammation	[14,17,32]
Retinopathy of Prematurity	Disrupted vessel growth, permeability changes	Retinal detachment, blindness
Hypertensive Retinopathy	Vessel wall thickening, leakage	Ischemia, microaneurysms, vision threat
Age-Related Macular Degeneration	Impaired perfusion, inflammatory activation	Drusen formation, atrophy, neovascular AMD

## Data Availability

No new data were created or analyzed in this study. Data sharing is not applicable to this article.

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
