# Peer review of "Disease Diagnosis Using Retinal Vasculature: Insights from Flammer Syndrome and AI"

_brainsci, 2025, doi:10.3390/brainsci15090919_

Round 1

Reviewer 1 Report

Comments and Suggestions for Authors

The review "Disease Diagnosis using Retinal Vasculature: Insights from Flammer Syndrome and AI" explores how retinal imaging can reveal systemic and ocular vascular dysfunction, building on Flammer’s work. It discusses the role of B vitamin therapy in improving retinal and lymphatic microcirculation, and highlights how AI can detect conditions such as glaucoma, autism, and potentially Alzheimer’s disease through retinal patterns. However, some revisions are needed, including stronger citations and more technical detail on the AI methodologies.

Major changes required:

  1. Lines 97–100: These lines reference a presentation given by Dr. Flammer in March 2025 on oxidative stress. While this is a valuable inclusion, I recommend strengthening the argument by incorporating additional references - ideally from peer-reviewed publications by Dr. Flammer and other researchers - to reinforce the claim that Flammer Syndrome is highly relevant to the understanding and management of normal tension glaucoma.
  2. Figure 1: As the only figure in the review and intended to summarize the effects of vascular dysfunction, I believe it would benefit from reorganization to include the role of retinal endothelial cells. I suggest structuring the schematic by first highlighting their main functions, followed by two separate panels: one illustrating the pathological outcomes and the other the upstream triggers. These triggers should be linked to therapeutic strategies, while the pathological outcomes should be connected to broader systemic implications.
  3. AI section: This section feels rather concise, especially considering the title of the review. I recommend expanding on the AI technologies used, briefly describing the types of algorithms (e.g., deep learning, convolutional neural networks) and the parameters analyzed. It would also be helpful to elaborate on the clinical implications, such as early screening, improved diagnostic accessibility, and monitoring of neurodegenerative diseases.

Minor changes required:

  1. Repetitive phrasing: As noted above, lines 23–24 and 35–36 contain similar ideas. I suggest merging them to improve flow and avoid redundancy. Please also check for similar instances throughout the manuscript.
  2. Lines 272–275: The author cites studies that highlight an association between retinal pigmentation and self-reported ethnicity. To further strengthen this statement, it would be helpful to refer, where possible, to additional specific studies published in peer-reviewed journals.

Author Response

Thank you for these helpful recommendations. I have addressed the points as follows. The major changes are modified in the following manner:

  1. (lines 97-100 and following to 105). I agree and have added references to Flammer and other works to build the referencing to published works on normal tension glaucoma and Flammer syndrome (in addition to the presentation by Dr Flammer, which is retained)
  2. I quite like your suggestion for the figure, and have incorporated it in full. The figure now has 2 additional panels that summarize causes of damage to endothelial cells that lead to vascular dysfunction.
  3. The first sentence of this section is now a paragraph that reviews a number of published works using AI systems to detect glaucoma and what parameters are involved. This paragraph is lines 253-264.

The minor points are

  1. The repetition has been merged (lines 23-27)
  2. I now refer to a useful study in debiasing AI diagnosis in diabetic retinopathy to address this concern. (lines 299-302)

Reviewer 2 Report

Comments and Suggestions for Authors

The author summarized a very important phenomenon in the field of ophthalmology, namely Flammer Syndrome. The review is well-written and cover many insightful topics for the relationship between vessels and health. I still have 2 main comments: 

- The title of the review clearly indicates the use of AI insights in retinal vasculature. but this was superficially presented in the last section of the review. I suggest to include wider score of AI part.

- Further, there is lack of citing other established and direct techniques to scrutinize retinal vasculature dysregulation such as OCTA, dynamic vessel analyzer. 

Author Response

Thank you for your kind words about this field, and for your recommendations. I have now expanded the last section to directly address your 2 main comments. I now open this section with a paragraph of other established AI techniques, where I discuss a range of AI systems used in detection of glaucoma and the parameters that are evaluated. This is lines 253-264.

Round 2

Reviewer 1 Report

Comments and Suggestions for Authors

The author has made the requested changes, which in my opinion have improved the overall quality of the manuscript.